# PROCEEDINGS A

statistical physics, complexity

ecological public goods, cooperation, social dilemma, environmental feedback, switching control

**Author for correspondence:**
Feng Fu
e-mail: feng.fu@dartmouth.edu

# Steering eco-evolutionary game dynamics with manifold control

Xin Wang[1,2], Zhiming Zheng[1] and Feng Fu[2,3]

[1]LMIB, NLSDE, BDBC, PCL and School of Mathematical Sciences, Beihang University, Beijing 100191, People's Republic of China
[2]Department of Mathematics, Dartmouth College, Hanover, NH 03755, USA
[3]Department of Biomedical Data Science, Dartmouth College, Lebanon, NH 03756, USA

XW, 0000-0001-6825-1031; FF, 0000-0001-8252-1990

Feedback loops between population dynamics of individuals and their ecological environment are ubiquitously found in nature and have shown profound effects on the resulting eco-evolutionary dynamics. By incorporating linear environmental feedback law into the replicator dynamics of two-player games, recent theoretical studies have shed light on understanding the oscillating dynamics of the social dilemma. However, the detailed effects of more general *nonlinear* feedback loops in multi-player games, which are more common especially in microbial systems, remain unclear. Here, we focus on ecological public goods games with environmental feedbacks driven by a nonlinear selection gradient. Unlike previous models, multiple segments of stable and unstable equilibrium *manifolds* can emerge from the population dynamical systems. We find that a larger relative asymmetrical feedback speed for group interactions centred on cooperators not only accelerates the convergence of stable manifolds but also increases the attraction basin of these stable manifolds. Furthermore, our work offers an innovative manifold control approach: by designing appropriate switching control laws, we are able to steer the eco-evolutionary dynamics to any desired population state. Our mathematical framework is an important generalization and complement to coevolutionary game dynamics, and also fills the theoretical gap in guiding the widespread problem of population state control in microbial experiments.

# 1. Introduction

The feedback between environment and evolutionary dynamics is widespread in a large number of natural systems [1–5]. In human society, a depleted environment or resource state favours cooperation, which results in the mutual growth of both the environmental state and cooperators, while subsequently free-riders increase, which in turn leads to the degradation of the environment [6,7]. Understanding this oscillating system dynamics is of vital importance when dealing with worldwide problems in human society, ranging from overgrazing of common pasture land overfishing to some big challenges such as pollution control and global warming [8–12]. Similar joint effects can also be obtained in some psychological–economic systems, such as social welfare, overuse of antibiotics and anti-vaccine problems [13–16].

In particular, similar eco-evolutionary feedback loops exist broadly in microbial systems, and these have received substantial prior attention in evolutionary biology and systems biology in recent years [17–20]. Among microbes, cooperation often emerges as a result of the secretion or the release of public goods, such as extracellular enzymes or extracellular antibiotic compounds [21–25]. A fundamental problem is how these bidirectional feedbacks between ecology and evolutionary dynamics affect the emergence of long-term cooperation as well as the corresponding ecological consequences [20,26–30]. An experimental study proves that the existence of the coupled interaction between population and evolutionary dynamics determines the demographic fate of social microbial populations [31]. Besides population density, there are many other ecological properties that have been found to play an important role in such reciprocal feedbacks, such as spatial structures of the population, resource regeneration and supply capacity [4,5,32–34].

Furthermore, to give a clear insight into these feedback-evolving games, a theoretical framework called coevolutionary game theory is proposed to analyse the coupled evolution of strategies and environment [35–39]. The core idea is to incorporate the game–environment feedback mechanism into replicator dynamics, in which the feedback changes the pay-off structure and further influences the evolution of strategies [40,41]. Such a framework successfully shows the emergence of an oscillating tragedy of the commons. Similar cycles are also confirmed in asymmetric evolutionary games with heterogeneous environments [42]. As a meaningful example of the application, the framework is used to explore the effects of the intrinsic growing capacity of the resources with punishment and inspection mechanisms [43]. In summary, these eco-evolutionary models reveal the great role that the feedback loop plays in resolving social dilemma and promoting the emergence of long-term cooperation.

However, it has been proved that, in most microbial systems, the essential factor that creates density-dependent (or other ecological property-dependent) selection, which leads to the existence of a feedback loop, is the preferential access to the common good for cooperators [44,45]. In other words, the preferential access mechanism affects the welfare of cooperators and defectors, which further determines the selection direction of ecological properties. This indicates an important fact that the current reward and pay-off structures of cooperators in a public goods game (PGG) may actually be influenced by how well they fare against defectors (namely, the natural selection gradient in the population). More specifically, the multiplication factor of cooperators may change during evolution. The changes depend directly on the current pay-offs of cooperators and defectors. This process in turn affects the evolutionary dynamics and leads to the existence of an asymmetrical feedback.

While most of the previous works focus on two-player games with environmental feedback [40,42], this selection gradient engineering via asymmetrical feedback in PGG, which is more general in microbial systems, is ignored; the effects of this feedback loop remain unclear. In addition, there is still a lack of proper understanding of nonlinear feedback mechanisms since most studies concentrate on linear feedback laws. Further, when looking into the current experimental results in microbial systems, we find that there exists a big theoretical gap in how to effectively steer a given initial population state to a desired final state with external feedback

control laws in coevolutionary games dynamics, which may have wide applications in systems biology.

To advance all these important issues, in this work we propose a general framework which extends the two-player games with environmental feedback to coevolutionary multi-player games with asymmetrical feedback driven by a nonlinear selection gradient. Unlike the solely interior equilibrium phenomenon obtained in previous models, we find the emergence of multiple segments of stable and unstable manifolds with a number of different feedback control functions. It is noteworthy that the equilibrium curve of an unstable manifold circumstance in our model is in line with the separatrix obtained by experimental results in [31]. In addition, a larger relative changing speed of the asymmetrical feedback can not only accelerate the convergence speed of stable manifolds but also increase the attraction basin of these stable manifolds. Our model reveals the detailed effects of the asymmetrical feedback loop driven by a nonlinear selection gradient in PGG, which is an important complement and generalization for the previous coevolutionary framework.

Moreover, we develop a new manifold control method based on our framework: when incorporating time-dependent or state-dependent switching laws that can actually control the stability of the possible manifolds [46–48], we can steer the coevolutionary games dynamics to any desired region. Therefore, our framework can be widely applied into culture refresh modes for establishing continuous culture devices and designing the required chemostat for microbial experiments, which is of great significance in systems biology and microbial ecology [49–51].

## 2. Modelling framework

Here, we propose a general framework for evolutionary dynamics with feedback loops in PGG, which is based on the current studies on two-player coevolutionary game theory [40]. We begin from a further observation of the experimental results in [31], which confirms the existence of a strong feedback loop between laboratory yeast population dynamics and the evolutionary dynamics of cooperators, the *SUC2* gene, as shown in figure 1*a*. The phase graph of the coevolutionary system is clearly divided into two regions by the separatrix line, above which the system converges to an eco-evolutionary equilibrium and the population survives and below which the population goes extinct. Of particular interest, a specific bi-phasic logistic mathematical model based on the Lotka–Volterra model of competition has been proposed to reproduce the emergence of the separatrix line in [31], in which the population growth rates of cooperators and defectors are distinguished under different cooperator densities. Although originating from the laboratory yeast, these experimental results are representative of the commonplace existence of feedback loops in many natural microbial systems. However, for a better understanding of all those similar coevolutionary dynamics of population strategy and ecological properties, a general framework is needed such that the population density in figure 1*a* can be included as an example. Further, in figure 1*b*, we raise an important control problem in general coevolutionary games dynamics, which has wide applications in similar microbial experiments: how can we effectively steer the given initial population state $(x_0, y_0)$ to a desired final state $(x_1, y_1)$ with external feedback control laws? To make progress on all these unsolved issues, in what follows we present a novel coevolutionary model with feedback laws that can be engineered based on the nonlinear selection gradient in the system.

Consider a well-mixed population. An individual finds itself in a group of size $S + 1$, with $S$ other players participating in the PGG. All players can choose to be either a cooperator that contributes $c$ to the public pool or a defector that free rides others' efforts in the group. In classical PGG, the total contributions are multiplied by a multiplication factor $r$ and then divided equally among all participants. In this work, to describe the phenomenon that cooperators have preferential access to the common good in real microbial systems, which facilitates the formation

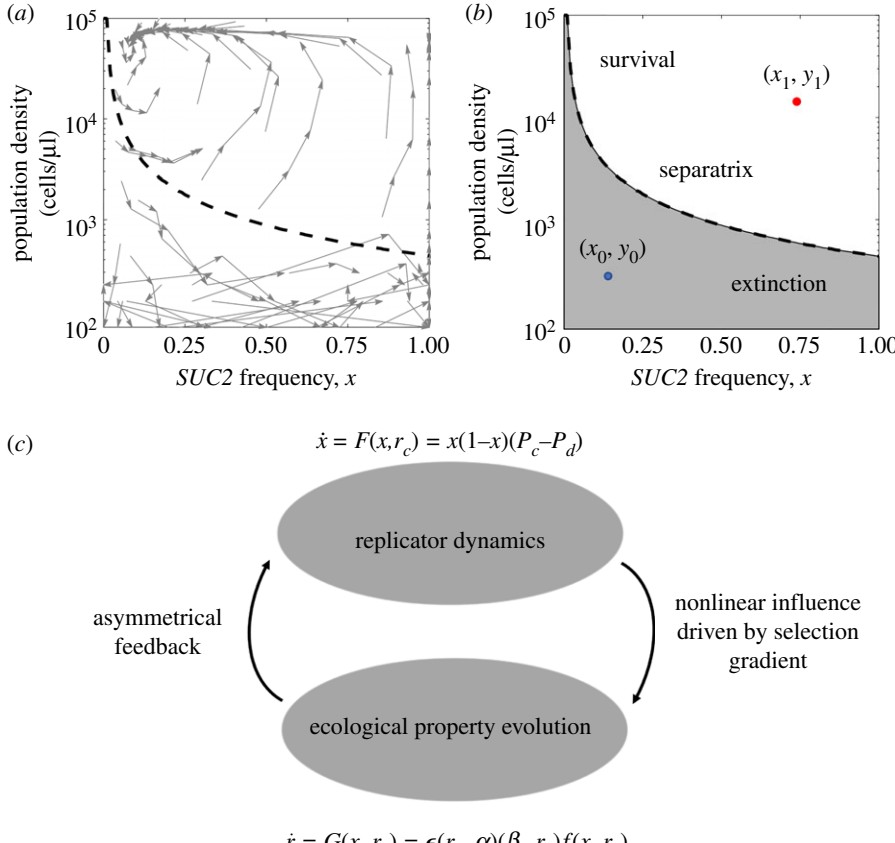

**Figure 1.** Experiment results in [31] and schematic of our eco-evolutionary model framework. (*a*) Eco-evolutionary trajectories of yeast population density and the fraction of cooperators, *SUC2* gene. This experimental result consists of 60 cultures over five growth–dilution cycles. The phase graph shows clearly two regions divided by a separatrix line, above which the system converges to an eco-evolutionary equilibrium and below which the population goes extinct. The separatrix line is predicted by a bi-phasic logistic model, provided in [31]. (*b*) Theoretical gap in the control problems of coevolutionary dynamics in microbial systems: how to steer the given initial population state $(x_0, y_0)$ to a desired final state $(x_1, y_1)$ with external feedback control laws. (*c*) Schematic of the model framework: eco-evolutionary games with asymmetrical feedback driven by a nonlinear selection gradient in a PGG.

of an asymmetrical feedback loop, we assume that the multiplication factors of cooperators and defectors are different, denoted by $r_c$ and $r_d$, respectively, and $r_c \geq r_d$. Without loss of generality, we keep $r_d$ constant and let $r_c$ change based on nonlinear feedback driven by the selection gradient (i.e. the nonlinear control function of the current pay-offs of cooperators and defectors) [52]. In turn, $r_c$ affects the relationship of the cooperators' and defectors' pay-offs and further drives evolutionary dynamics. In this way, we characterize all kinds of ecological properties affected by the fraction of cooperators as a benefit of cooperation, which can be described by the multiplication factor of cooperators. The schematic of this coevolutionary game framework is shown in figure 1*c*.

Assume that $x$ denotes the frequency of cooperators in the population. For any given focal individual, the chance that $k$ out of other $S$ individuals are cooperators is

$$\binom{S}{k} x^k (1-x)^{S-k}.$$

For simplicity and without loss of generality, we set the cooperator's cost $c$ equal to 1. Therefore, the expected pay-offs of cooperators and defectors, $P_c$ and $P_d$, are

$$
\begin{aligned}
P_c &= \sum_{k=0}^{S} \binom{S}{k} x^k (1-x)^{S-k} \left[ \frac{(k+1)r_c}{S+1} - 1 \right] \\
&= \frac{1+Sx}{S+1} r_c - 1 \\
P_d &= \sum_{k=0}^{S} \binom{S}{k} x^k (1-x)^{S-k} \frac{kr_d}{S+1} \\
&= \frac{Sx}{S+1} r_d.
\end{aligned}
\tag{2.1}
$$

and

Then the replicator dynamics for the fraction of cooperators $x$ is

$$
\begin{aligned}
\dot{x} &= x(P_c - \bar{P}) \\
&= x(1-x)(P_c(x, r_c) - P_d(x, r_d)) \\
&= x(1-x)\left( \frac{Sx+1}{S+1} r_c - 1 - \frac{Sx}{S+1} r_d \right),
\end{aligned}
\tag{2.2}
$$

where $\bar{P} = xP_c + (1-x)P_d$ represents the average pay-off of the population. Meanwhile, the feedback-evolving dynamics, i.e. the ecological property evolution in figure 1c, is given by

$$
\dot{r}_c = \epsilon(r_c - \alpha)(\beta - r_c)f(x, r_c),
\tag{2.3}
$$

where $\epsilon \geq 0$ denotes the relative changing speed of $r_c$, the cooperator's multiplication factor, compared with strategy dynamics. The logistic term $(r_c - \alpha)(\beta - r_c)$ ensures that $r_c$ is restrained to the range $[\alpha, \beta]$, which satisfies $1 < \alpha < \beta < S+1$ according to the social dilemma in PGG. In addition, $f(x, r_c)$ is a control function that describes the asymmetrical feedback mechanisms in our model. While $f$ actually characterizes the current impact of population strategies on the environment, previous works exclusively focus on linear selection gradient feedback laws, such as $f = \theta x - (1-x)$ in [40] and $f = e_L x + e_H(1-x)$ in [41]. Here, we stress our effects on the generality of nonlinearity in feedback control laws, the general form of which is

$$
f(x, r_c) = \Phi_0(x, r_c)(\Phi_1(x, r_c) - a_1)(\Phi_2(x, r_c) - a_2) \cdots (a_n - \Phi_n(x, r_c)),
\tag{2.4}
$$

in which

$$
\begin{aligned}
\Phi_0(x, r_c) &= P_c(x, r_c) - P_d(x, r_d) \\
\Phi_i(x, r_c) &= \theta_i P_c(x, r_c) - P_d(x, r_d)
\end{aligned}
\tag{2.5}
$$

and

and $a_i \geq 0$, $\theta_i > 0$, $n+1$ denotes the order of the control function. Here $\Phi_0(x, r_c) = P_c - P_d$ is the simplest form of the linear selection gradient. Therefore, equation (2.4) provides a series of control functions driven by the nonlinear selection gradient in a general polynomial form. Also note that, to construct a stabilizing selection law, the sign of the last term (i.e. $(a_n - \Phi_n(x, r_c))$) in equation (2.4) is contrary to the other terms [52].

Finally, our generalized framework of multi-player evolutionary games with asymmetrical feedback driven by nonlinear selection gradient can be written as follows:

$$
\begin{aligned}
\dot{x} &= x(1-x)(P_c - P_d) \\
\dot{r}_c &= \epsilon(r_c - \alpha)(\beta - r_c)(P_c - P_d)((\theta_1 P_c - P_d) - a_1) \cdots (a_n - (\theta_n P_c - P_d)).
\end{aligned}
\tag{2.6}
$$

and

# 3. Results

## (a) Emergence of multiple segments of stable and unstable equilibrium manifolds

Firstly, we analyse the simplest circumstance in which the feedback control term $f$ is a quadratic function, i.e. $n = 1$,

$$f(x, r_c) = (P_c - P_d)(a_1 - (\theta_1 P_c - P_d)). \tag{3.1}$$

The co-evolutionary model is explicitly described by

$$\left.\begin{aligned} \dot{x} &= x(1 - x)(P_c - P_d) \\ \dot{r}_c &= \epsilon(r_c - \alpha)(\beta - r_c)(P_c - P_d)(a_1 - (\theta_1 P_c - P_d)). \end{aligned}\right\} \tag{3.2}$$

and

In figure 2, we show the emergence of multiple segments of stable and unstable equilibrium manifolds using phase graphs under different conditions. For simplicity and without loss of generality, the parameters are chosen as follows: $\alpha = 1.5$, $\beta = 3.5$, $S = 3$, $r_d = 1.5$, $\epsilon = 2$, $\theta_1 = 2$ and $a_1 = 2, 0.5, 0$. In addition, in figure 2d, we present all the curves that satisfy $f = 0$ in figure 2a–c. In the following parts of this paper, to clearly describe the stability of fixed points, we name $P_c = P_d$ as the equilibrium curve and $\theta_i P_c - P_d = a_i$ as the control curves. Under these parameters, there are five possible fixed points on the boundary and an equilibrium curve for the system in total. Among the five boundary fixed points, $(x = 0, r_c = 1.5)$ is always stable and $(x = 0, r_c = 3.5)$, $(x = 1, r_c = 1.5)$, $(x = 1, r_c = 3.5)$ are always unstable, while the stability of $(x = 1, r_c = (25/16) + (a_i/2))$ depends on the parameter $a_i$. Detailed proofs are provided in electronic supplementary material, appendix A. Of particular interest, here we focus on the stability of the equilibrium curve. Unusually, we obtain the emergence of stable equilibrium manifolds in both figure 2b,c, which means that the system can finally evolve to many different stable states depending on the initial conditions. This new phenomenon is totally different from the solely interior fixed points situation discussed in previous models [40,41]. Moreover, the equilibrium curve of unstable manifold situations in our model (figure 2a) is in line with the phase separatrix observed by experimental work in figure 1a, which indicates the potential power for our general framework to explain the abundant eco-evolutionary phenomena shown in real microbial systems. Additionally, we provide detailed proof of the stability of the manifolds in the following section. For simplicity and without loss of generality, we use parameters given in figure 2c in which $\theta_1 = 2$ and $a_1 = 0$ as an example. The schematic of this proof is shown in figure 3.

Assume $(x^*, r_c^*)$ is an arbitrary fixed point on the equilibrium curve $P_c = P_d$. The phase plane is separated into two regions by the equilibrium curve, above which is called region $A$ and below which is region $B$. Then $(x^*, r_c^*)$ is stable if and only if

$$\boldsymbol{w_i} \cdot \boldsymbol{v_i} < 0, \tag{3.3}$$

in a small neighbourhood of $(x^*, r_c^*)$, where $\boldsymbol{v_i}$ is the normal vector of the equilibrium curve while $\boldsymbol{w_i}$ is the trajectory field direction, $i = 1, 2$. According to equation (2.1), the equilibrium curve $P_c = P_d$ can be written as

$$r_c = r_d + \frac{S + 1 - r_d}{Sx + 1}. \tag{3.4}$$

Defining $h(x)$ as the slope equation of the equilibrium curve, we have

$$h(x) = \frac{dr_c}{dx} = -\frac{S(S + 1 - r_d)}{(Sx + 1)^2}. \tag{3.5}$$

Therefore, we obtain the normal vectors of the equilibrium curve at $(x^*, r_c^*)$:

$$\left.\begin{aligned} \boldsymbol{v_1} &= (-h(x^*), 1) \\ \boldsymbol{v_2} &= (h(x^*), -1). \end{aligned}\right\} \tag{3.6}$$

and

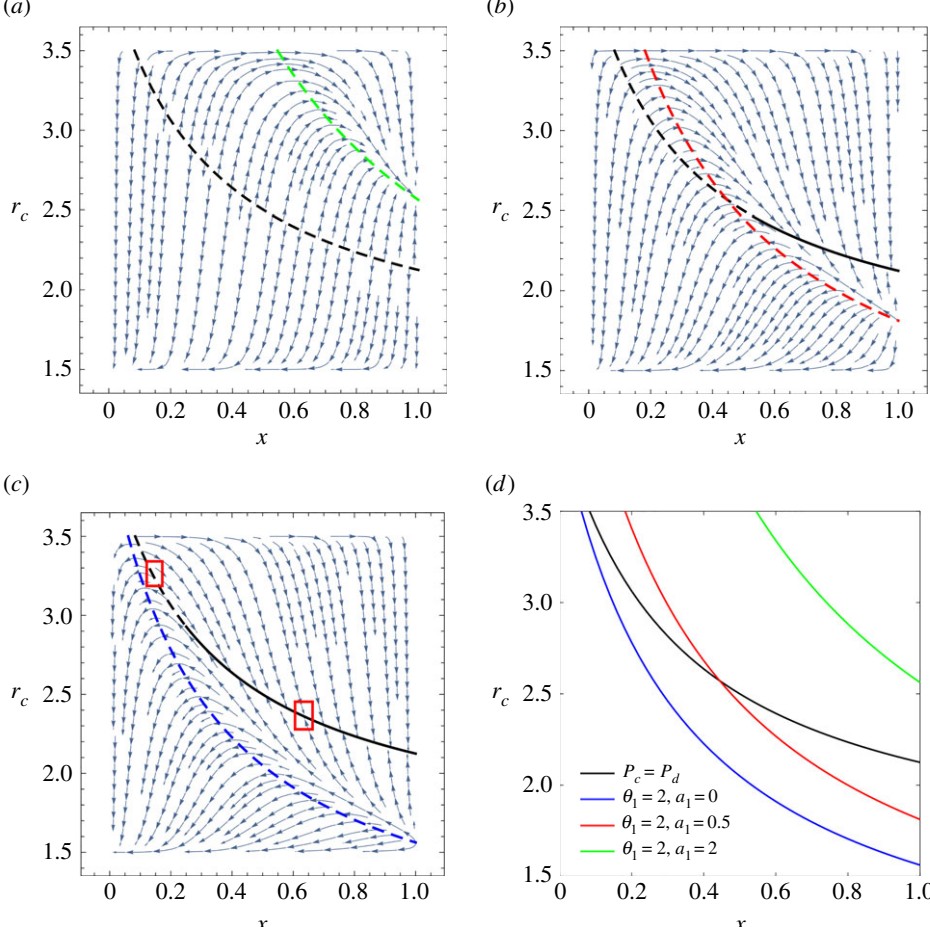

**Figure 2.** Emergence of stable equilibrium manifolds when the feedback control function $f$ is in quadratic form. ($a$–$c$) Phase graph under different control conditions. Throughout, $\alpha = 1.5$, $\beta = 3.5$, $S = 3$, $r_d = 1.5$, $\epsilon = 2$, $\theta_1 = 2$. We change $a_1 = 2, 0.5, 0$. The stable and unstable parts of the equilibrium curve are indicated by solid and dashed lines, respectively. In particular, in ($c$), we mark the typical stable and unstable manifold situations in a small neighbourhood of the equilibrium curve with rectangular boxes and ($d$) presents all curves that satisfy $f = 0$ in ($a$–$c$). (Online version in colour.)

On the other hand, the trajectory vector $(\dot{x}, \dot{r}_c)$ satisfies $\dot{r}_c < 0$, $\dot{x} > 0$ when $(x, r_c)$ is in region $A$ and $\dot{r}_c > 0$, $\dot{x} < 0$ when $(x, r_c)$ is in region $B$, according to equation (3.2). Thus, the trajectory field directions in a small neighbourhood of $(x^*, r_c^*)$ read

$$\left.\begin{array}{l} w_1 = (1, \lim_{P_c \to P_d} g(x^*)), P_c \to P_d + \delta \\[2mm] w_2 = (-1, -\lim_{P_c \to P_d} g(x^*)), P_c \to P_d - \delta, \end{array}\right\} \tag{3.7}$$

and

where $g(x)$ is the slope function of the trajectory field,

$$g(x) = \frac{\mathrm{d}r_c}{\mathrm{d}x} = \frac{\mathrm{d}r_c/\mathrm{d}t}{\mathrm{d}x/\mathrm{d}t}$$

$$= \frac{\epsilon(r_c - \alpha)(\beta - r_c)(P_c - P_d)(a_1 - (\theta_1 P_c - P_d))}{x(1 - x)(P_c - P_d)}. \tag{3.8}$$

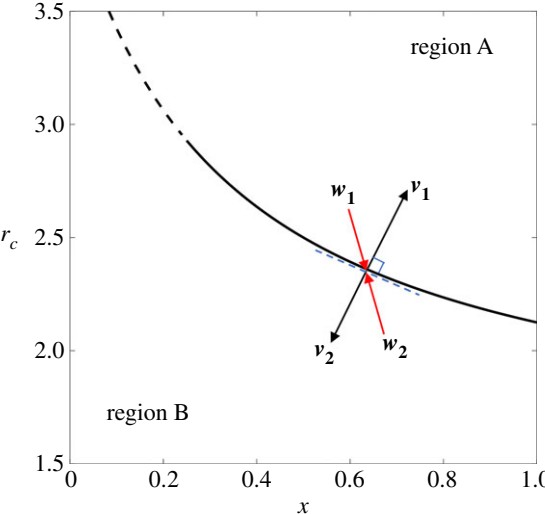

**Figure 3.** Schematic of the stability proof. The fixed point on the equilibrium curve is stable if and only if $w_i \cdot v_i < 0$ in a small neighbourhood, where $v_i$ is the normal vector of the equilibrium curve and $w_i$ is the trajectory field direction. (Online version in colour.)

Substituting equations (3.6) and (3.7) in equation (3.3), we reach the necessary and sufficient condition that $(x^*, r_c^*)$ is stable,

$$\lim_{P_c \to P_d} g(x^*) < h(x^*). \tag{3.9}$$

Finally, let $\alpha = 1.5$, $\beta = 3.5$, $S = 3$, $r_d = 1.5$, $\epsilon = 2$, $\theta_1 = 2$ and $a_1 = 0$; then we have

$$\left.\begin{aligned} h(x^*) &= -\frac{7.5}{(3x^* + 1)^2} \\ \text{and} \qquad \lim_{P_c \to P_d} g(x^*) &= \frac{540x^* - 45}{16(3x^* + 1)^2(x^* - 1)}. \end{aligned}\right\} \tag{3.10}$$

Note that $r_c \in [1.5, 3.5]$, which leads to $x \in [1/12, 1]$ according to equation (3.4). Substituting equations (3.10) in equation (3.9), we have $x^* > 0.25$. Therefore, when $x > 0.25$, the trajectories in the neighbourhood of the fixed points will converge to the equilibrium curve, which proves the stability of the manifolds. On the contrary, when $x \in [1/12, 0.25)$, we have $w_i \cdot v_i > 0$, and the trajectories in the neighbourhood evolve away from the equilibrium curve, a situation in which the manifolds are unstable. Therefore, $x = 0.25$ is actually a saddle point of the system which satisfies $w_i \cdot v_i = 0$. The typical stable and unstable manifold situations in a small neighbourhood of the equilibrium curve are marked with rectangular boxes in figure 2c. Similarly, we derive that the stable region of the equilibrium curve is $x \in (0.5595, 1)$ in figure 2b, while there only exist unstable manifolds in figure 2a. In summary, we have proved the emergence of stable equilibrium manifolds in our framework. Meanwhile, we provide detailed approach to calculate the saddle point of the system, which is the critical point for the stability of the equilibrium curve.

In figure 4, we present the detailed effects of the relative asymmetrical feedback speed. Throughout the figure, $\alpha = 1.5$, $\beta = 3.5$, $S = 3$, $r_d = 1.5$, $\theta_1 = 4$, $a_1 = 0$. We give $\epsilon = 0.1, 0.5, 2$ in figure 4a–c, respectively. According to the results presented in the first column, which shows the phase graphs under different circumstances, we find that the relative changing speed of the cooperator's multiplication factor has a strong influence on the slope of the trajectories and further affects the position of the saddle point on the equilibrium curve, i.e. it influences the stability of the manifolds. When the relative feedback speed $\epsilon$ is larger, the trajectory field is steeper and the attraction basin of these stable manifolds increases. We prove this conclusion analytically using

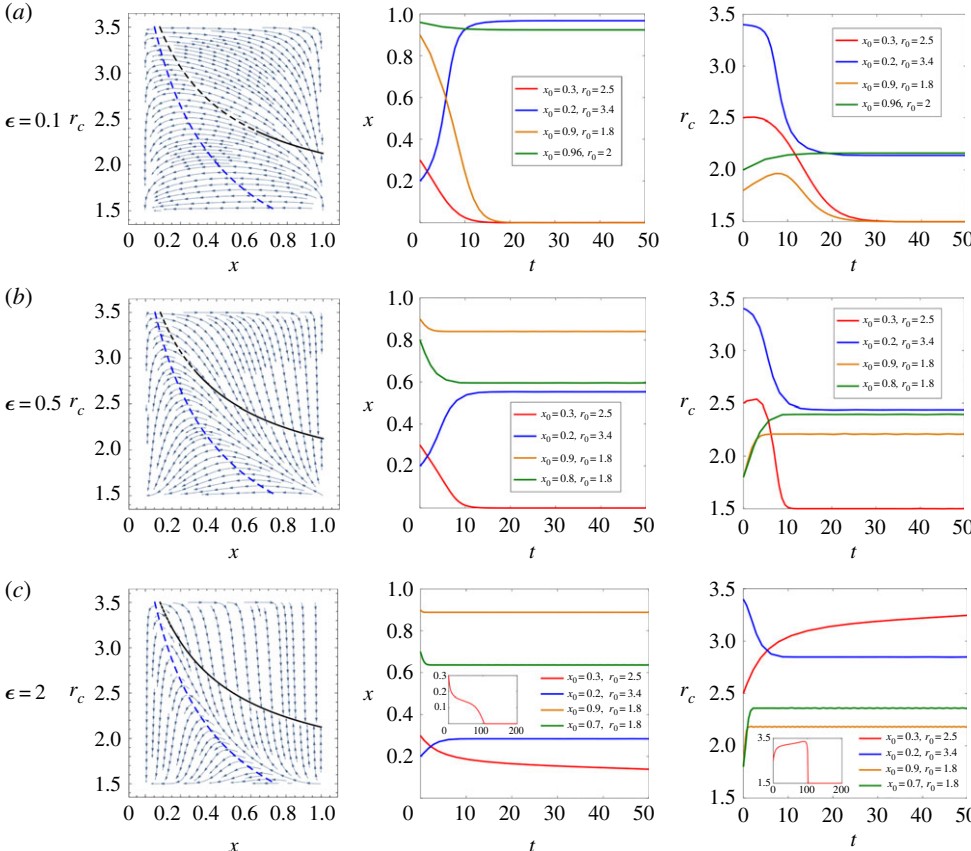

**Figure 4.** The effects of relative asymmetrical feedback speed. Throughout, $\alpha = 1.5$, $\beta = 3.5$, $S = 3$, $r_d = 1.5$, $\theta_1 = 4$, $a_1 = 0$. We change $\epsilon = 0.1, 0.5, 2$ in $(a–c)$, respectively. The first column presents phase graphs in different situations while the second and third columns give variations of strategy dynamics as well as the cooperator's multiplication factor over time, correspondingly. We use solid versus dashed lines to indicate the stability of the equilibrium curve in all phase graphs. (Online version in colour.)

the same approach as shown in the stability proof section. We keep $\epsilon$ as a variable and the other parameters are the same as in figure 4.

Restraining the slope function of the trajectory field $g(x)$ in a small neighbourhood of the equilibrium curve, i.e. letting $P_c \to P_d$, leads to $r_c \to 1.5 + 2.5/(3x + 1)$ and $x \in [1/12, 1]$,

$$\lim_{P_c \to P_d} g(x) = \frac{\epsilon(r_c - \alpha)(\beta - r_c)(a_1 - (\theta_1 P_c - P_d))}{x(1 - x)}$$

$$= \frac{135\epsilon(12x - 1)}{32(3x + 1)^2(x - 1)}. \tag{3.11}$$

Let $\lim_{P_c \to P_d} g(x) = h(x)$; then we have

$$x^*(\epsilon) = \frac{27\epsilon + 48}{324\epsilon + 48}$$

$$= \frac{1}{12} + \frac{44}{324\epsilon + 48}. \tag{3.12}$$

Therefore, when $x > x^*$ the equilibrium curve is stable, while when $x \in [1/12, x^*)$ the equilibrium curve is unstable. Finally, note that $x^*(\epsilon)$ is a monotone decreasing function which indicates that, when $\epsilon$ is larger, $x^*$ becomes smaller, i.e. the attraction basin of these stable manifolds increases.

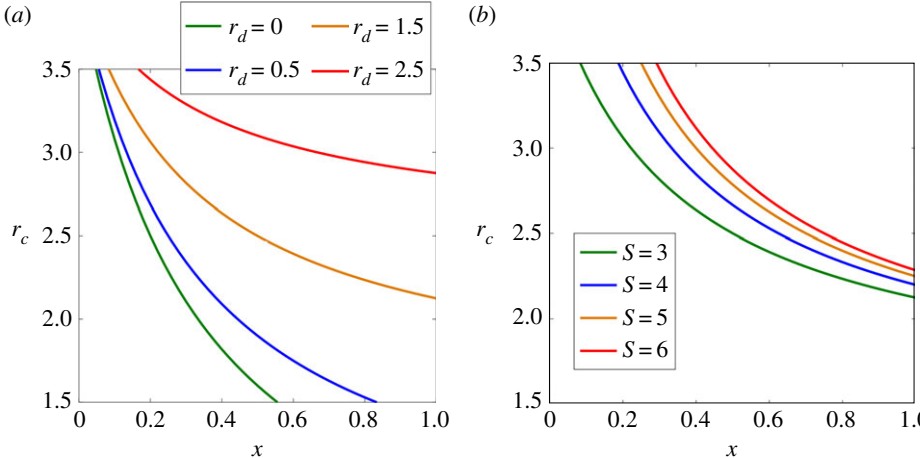

**Figure 5.** How the defector's multiplication factor $r_d$ and the group size $S$ influence the position of the equilibrium curve and stable manifolds. (a) We fix $S = 3$ and change $r_d = 0, 0.5, 1.5, 2.5$. (b) We fix $r_d = 1.5$ and set $S = 3, 4, 5, 6$. (Online version in colour.)

In addition, the second and third columns of figure 4 give variations of the strategy dynamics and the cooperator's multiplication factor over time, respectively. In each subfigure, we present four different initial conditions. Our results show that a larger $\epsilon$ effectively accelerates the convergence speed of the stable manifold, but it may actually slow down the converging process of unstable manifolds towards the mutual defection state.

In figure 5, we show the determinants of the position of the equilibrium curve as well as their detailed effects. In the general framework, the equilibrium equation $r_c = r_d + (S + 1 - r_d)/(Sx + 1)$ indicates that the position of the equilibrium curve is determined by the defector's multiplication factor $r_d$ and the group size $S$, according to equation (3.4). Moreover, we have

$$\left.\begin{aligned} \frac{\partial r_c}{\partial r_d} &= \frac{Sx}{Sx + 1} \\ \frac{\partial r_c}{\partial S} &= \frac{1 - (1 - r_d)x}{(Sx + 1)^2} \end{aligned}\right\} \tag{3.13}$$

and

in which $r_d \geq 0$ and $x \in [0, 1]$. Therefore $(\partial r_c/\partial r_d) \geq 0$, $(\partial r_c/\partial S) \geq 0$, which means that $r_c$ increases as $r_d$ or $S$ becomes larger. We present these varying trends in figure 5 in detail, in which $S = 3$ and $r_d = 0, 0.5, 1.5, 2.5$ in figure 5a while $r_d = 1.5$ and $S = 3, 4, 5, 6$ in figure 5b. It is interesting to notice the similarities and connections between our theoretical outcomes and the experimental results in the yeast case. Sanchez & Gore [31] further create three environments using different dilution factors: a 'benign' environment with a low dilution factor, an 'intermediate' environment with a moderate dilution factor and a 'harsh' environment with a high dilution factor. They found that the position of the separatrix line moves upwards as the environment becomes harsher. The same trend can be observed in our results (see figure 5a): when $r_d$ becomes larger, the relative pay-off of defectors increases and the cooperation environment gets worse; the equilibrium curves also moves upwards. This indicates that the relative value of $r_d$ in our model could be a simple symbol of the environmental state in real microbial systems.

Furthermore, in figure 6, we propose a more complex situation where $n = 2$ and $f$ is a cubic function, which can be written as

$$f(x, r_c) = (P_c - P_d)((\theta_1 P_c - P_d) - a_1)(a_2 - (\theta_2 P_c - P_d)). \tag{3.14}$$

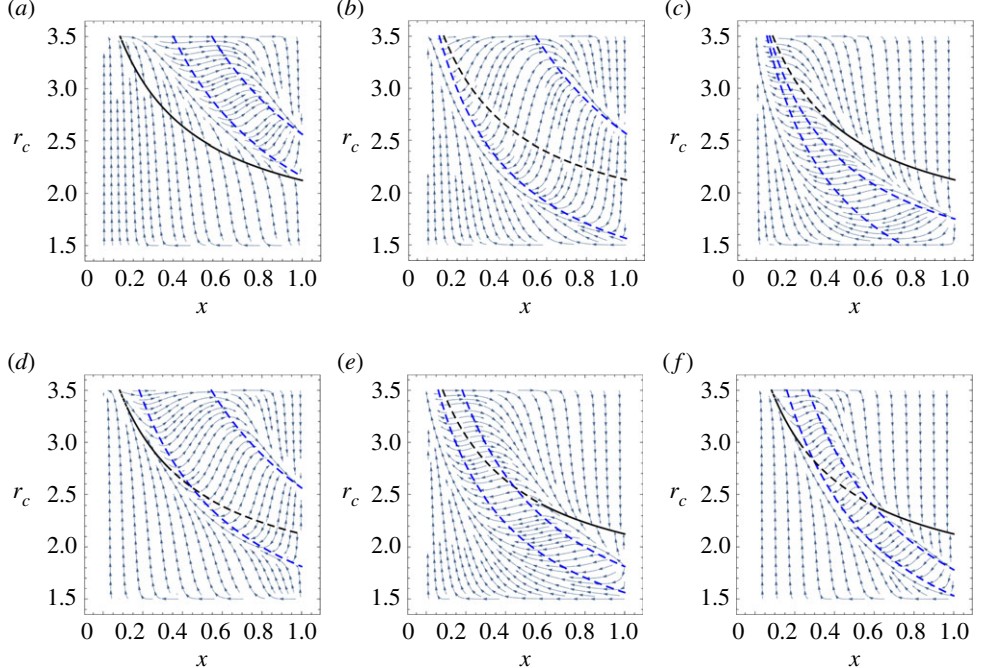

**Figure 6.** Phase graphs under different circumstances where the control function $f$ is in cubic form. Throughout, $\alpha = 1.5$, $\beta = 3.5$, $S = 3$, $r_d = 1.5$, $\epsilon = 2$. We use solid versus dashed lines to indicate the stability of equilibrium curves. (a) $\theta_1 = 2$, $a_1 = 1.2$, $\theta_2 = 2$, $a_2 = 2$. The stable range of the equilibrium curve is $x \in (0.09355, 1]$. (b) $\theta_1 = 2$, $a_1 = 0$, $\theta_2 = 2$, $a_2 = 2$. There only exist unstable manifolds. (c) $\theta_1 = 1.5$, $a_1 = 0$, $\theta_2 = 4$, $a_2 = 0$. The stable range of the equilibrium curve is $x \in (0.3395, 1]$. (d) $\theta_1 = 2$, $a_1 = 0.5$, $\theta_2 = 2$, $a_2 = 2$. The stable range of the equilibrium curve is $x \in (0.1237, 0.3198]$. (e) $\theta_1 = 2$, $a_1 = 0$, $\theta_2 = 2$, $a_2 = 0.5$. The stable range of the equilibrium curve is $x \in (0.5984, 1]$. (f) $\theta_1 = 4$, $a_1 = 1$, $\theta_2 = 4$, $a_2 = 2$. The stable range of the equilibrium curve is $x \in (0.1057, 0.2226)$ and $x \in (0.622, 1]$. (Online version in colour.)

Throughout the figure, $\alpha = 1.5$, $\beta = 3.5$, $S = 3$, $r_d = 1.5$, $\epsilon = 2$. In figure 6a–c, we present different combinations of the two control curves when there is no intersection between the control curves and the equilibrium curve, while there exists one intersection in figure 6d,e and two intersections in figure 6f. Not surprisingly, the trajectory field reveals rich probabilities and abundant new variation trends, an important one of which is that ($x = 0$, $r_c = 3.5$) becomes a stable fixed point while ($x = 0$, $r_c = 1.5$) becomes unstable, indicating an opposite path direction for a range of initial conditions compared with the $n = 1$ situations we discussed above. In addition, in figure 6a,c–f, we obtain stable equilibrium manifolds ranging from a small neighbourhood of the equilibrium curve, as in figure 6d, to a large region, as in figure 6a, as shown by the solid lines. On the other hand, figure 6b shows the completely unstable manifolds situation.

In general, our results reveal that we can control the stability of the equilibrium curve $P_c = P_d$ as well as the stability of the manifolds by giving different feedback control functions $f$ or changing the relative feedback speed $\epsilon$. This indicates the possibility for designing population-state-dependent switching control laws to steer the system evolution towards the desired directions based on our framework. We provide a detailed control approach as well as two control examples in the next section.

## (b) Manifold control with switching control laws

First, we incorporate two kinds of switching control laws in our general framework, one of which is time-dependent and the other of which is state-dependent.

The time dependent switching control law for the control function $f$ can be written as follows:

$$f_{\sigma(t)}(x, r_c) = f_{\sigma, k(\sigma)}(x, r_c)$$
$$= (P_c - P_d)((\theta_{\sigma,1}P_c - P_d) - a_{\sigma,1}) \cdots (a_{\sigma,k(\sigma)} - (\theta_{\sigma,k(\sigma)}P_c - P_d)). \qquad (3.15)$$

Here $\sigma(t) : [0, \infty) \to I$, where $I$ represents a finite index set $1, 2, \ldots, m$. In addition, $k(\sigma) + 1$ is the order of the control function $f_\sigma$.

Additionally, the state-dependent switching control law can be written as

$$f_s(x, r_c) = f_{s, k(s)}(x, r_c)$$
$$= (P_c - P_d)((\theta_{s,1}P_c - P_d) - a_{s,1}) \cdots (a_{s,k(s)} - (\theta_{s,k(s)}P_c - P_d)). \qquad (3.16)$$

Here $s : q \to I$, where $q \in Q$, which represents a family of switching surfaces/guards that partition the $x - r_c$ plant into finite regions, and $I$ is a finite index set $1, 2, \ldots, m$. $k(s) + 1$ is the order of the control function $f_s$.

In electronic supplementary material, appendix B, we provide a constructive proof for the existence of control laws when given a certain final state $(x_1, r_1)$ with an initial state $(x_0, r_0)$ in our general framework. We show the possibility of ending the evolution in any desired region, both above and below the equilibrium curve, with any fraction of final cooperators, as long as the initial fraction of cooperators $x_0$ is not too small, otherwise the evolution cannot even reach the equilibrium curve and directly ends in a mutual defection state. Note that our proof is a heuristic method for seeking a group of possible control functions, which might not be the only solution for a certain control problem.

For a better understanding, here we illustrate two detailed control examples in figure 7. The evolution begins from $(x_0, r_0) = (0.2, 1.6)$ and we give two desired final states: $(x_1, r_1) = (1, 2.5)$ and $(0.5, 2)$.

The first example only uses the time-dependent switching control law, in which the control functions and the relative feedback speeds are as follows:

$$\left. \begin{aligned} \epsilon_1 f_1 &= \epsilon_1 (P_c - P_d)((\theta_{1,1}P_c - P_d) - a_{1,1})(a_{1,2} - (\theta_{1,2}P_c - P_d)) \\ &= 2(P_c - P_d)((2P_c - P_d) - 1.2)(2 - (2P_c - P_d)) \\ \text{and} \quad \epsilon_2 f_2 &= \epsilon_2 (P_c - P_d)(a_{2,1} - (\theta_{2,1}P_c - P_d)) \\ &= 2(P_c - P_d)(1.875 - (2P_c - P_d)) \end{aligned} \right\} \qquad (3.17)$$

with $t$ large enough to end the evolutions at the stable fixed points. The evolution path is the blue trajectory followed by the red one, which finally stops at $(1, 2.5)$, as shown in figure 7a. Further, in figure 7b, we provide detailed time evolutions of the population state $(x, r_c)$ under two control stages. The colours of the curves correspond to the two $y$-axes while the types of the curves indicate the evolutions under different control laws. The colours of the control laws $\epsilon_i f_i$ correspond to the trajectories in figure 7a.

The second example is composed of three trajectories in figure 7a: the blue, green and orange ones, ending at $(0.5, 2)$. Here we use the time-dependent switching control law for the first two trajectories, and the state-dependent switching control law for the third control function to stop the evolution at $x = 0.5$. The control functions and the relative feedback speeds are as follows:

$$\left. \begin{aligned} \epsilon_1 f_1 &= 2(P_c - P_d)((2P_c - P_d) - 1.2)(2 - (2P_c - P_d)), \\ \epsilon_2 f_2 &= 2(P_c - P_d)((2P_c - P_d) - 0)(0.5 - (2P_c - P_d)) \\ \text{and} \quad \epsilon_3 f_3 &= 3.2(P_c - P_d)((2P_c - P_d) - 0)(2 - (2P_c - P_d)). \end{aligned} \right\} \qquad (3.18)$$

Additionally, in figure 7c, we show the time evolutions of the population state $(x, r_c)$ under three control stages. Specifically, control law 3 is state-dependent and the evolution stops at about $t = 203.8$.

In both examples, when the system evolves to the equilibrium curve and reaches a stable fixed point, we change the control function and meanwhile provide a small disturbance $\delta \leq 0.01$ to $r_c$

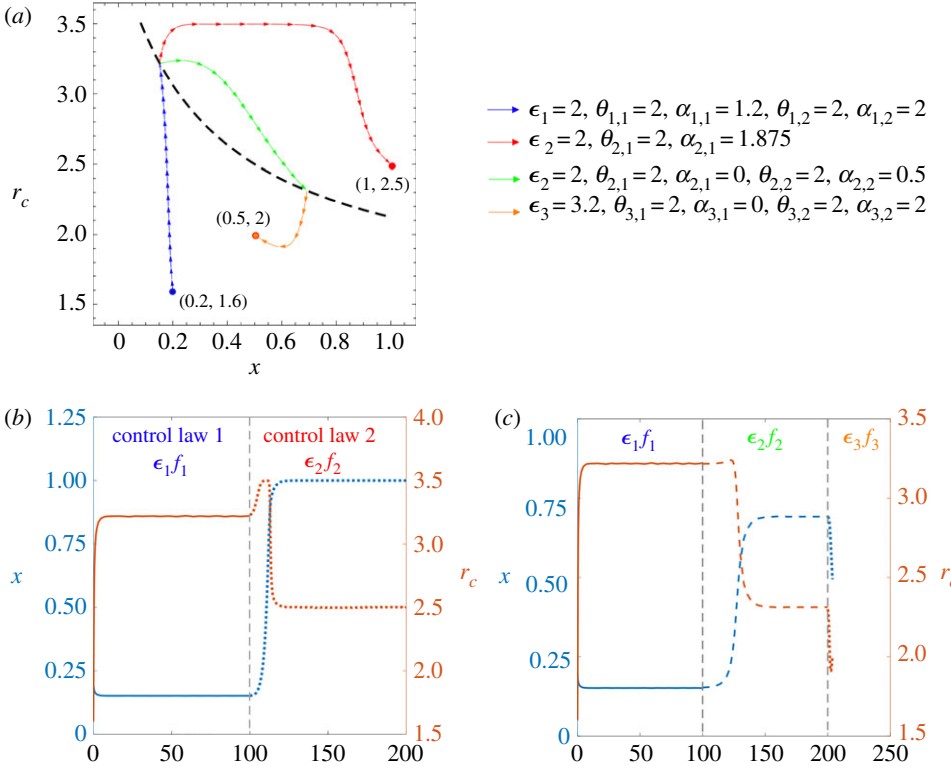

**Figure 7.** Two control examples with switching control laws in the general framework. The fixed parameters are as follows: $\alpha = 1.5, \beta = 3.5, S = 3, r_d = 1.5$. (a) Phase trajectories of two control examples. We begin from $(x_0, r_0) = (0.2, 1.6)$ and give two desired final states: $(x_1, r_1) = (1, 2.5)$ and $(0.5, 2)$. The first evolution path is the blue trajectory followed by the red one, ending at $(1, 2.5)$. Here we only use time-dependent switching control law, in which $\epsilon_1 = 2$ with $\theta_{1,1} = 2, a_{1,1} = 1.2, \theta_{1,2} = 2$, $a_{1,2} = 2$ for $f_1$ and $\epsilon_2 = 2$ with $\theta_{2,1} = 2, a_{2,1} = 1.875$ for $f_2$. Here $t$ is large enough to end the evolutions at the stable fixed points. The second example consists of three trajectories: the blue, green and orange ones, ending at $(0.5, 2)$. Here we use time-dependent switching control law for the first two trajectories, and state-dependent switching control law for the third control function to stop the evolution at $x = 0.5$. Detailed parameters for the relative feedback speeds and the control functions are as follows: $\epsilon_1 = 2$ with $\theta_{1,1} = 2, a_{1,1} = 1.2, \theta_{1,2} = 2, a_{1,2} = 2$ for $f_1$, $\epsilon_2 = 2$ with $\theta_{2,1} = 2, a_{2,1} = 0, \theta_{2,2} = 2, a_{2,2} = 0.5$ for $f_2$, and $\epsilon_3 = 3.2$ with $\theta_{3,1} = 2, a_{3,1} = 0, \theta_{3,2} = 2, a_{3,2} = 2$ for $f_3$. In both examples, when the system evolves to the equilibrium curve and reaches a stable fixed point, we change the control function and meanwhile provide a small disturbance $\delta \leq 0.01$ to $r_c$ according to the desired direction, by which means we restart the evolutions. (b,c) Detailed time evolutions of the population state $(x, r_c)$ in the first and second control examples, respectively. The colours of the curves correspond to the two $y$-axes while the types of the curves indicate evolutions under different control laws. The colours of control laws $\epsilon_i f_i$ correspond to the trajectories in (a). In particular, in (c), control law 3 is a state-dependent switching control law and the evolution ends at about $t = 203.8$.

according to the desired direction, which restarts the evolutions. The absolute errors for the final positions of $r_c$ when $x$ reaches $x_1$ are within 0.01, according to the numerical solutions of the equations.

Finally, we look back at the yeast experiment again in order to illustrate how our control method works in real experimental operations. According to Gore *et al.* [5], the cooperator strain in the experiments is a histidine auxotroph (i.e. one that cannot produce histidine endogenously). Consequently, the multiplication factor of the cooperators in our model can be modulated by controlling the histidine concentration in the growth medium; namely, the 'cost of cooperation',

which can change the growth rate of the cooperators relative to the defectors. This shows the practicability of our model framework to steer the coevoluntionary dynamics in real microbial systems.

## 4. Conclusion and discussions

The coevoluntionary game between environment and strategy dynamics has aroused great concern in recent years, because of the widespread existence of eco-evolutionary feedback loops and their great importance in a range of natural systems [53–55]. Despite the progress in two-player games with linear environmental feedback laws, a more general framework is required to describe the nonlinear coevoluntionary dynamics of the population strategy and population properties in PGG.

Inspired by the experimental results in [31] that confirm the existence of strong feedback loops as well as the emergence of the separatrix line in dynamics, we provide a modelling framework in multi-player games with asymmetrical feedback driven by a nonlinear selection gradient, taking into account the fact that the preferential access to the common good for cooperators directly brings about the emergence of feedback loops in most microbial systems. We consider ecological properties solely being affected by the fraction of cooperators, such as the population density in [31], as a benefit of cooperation that can be described by the cooperator's multiplication factor, which makes our framework more general. Unusually, we find multiple segments of stable and unstable equilibrium manifolds in phase graphs with a number of control functions, which is a new dynamical phenomenon that is totally different from the solely interior fixed equilibrium situation obtained in previous works. In addition, we show that the relative asymmetrical feedback speed for group interactions centred on cooperators can accelerate the converging process of the stable manifolds while slowing down the convergence speed of unstable manifolds. And a larger feedback speed increases the attraction basin of these stable manifolds. In addition, the position of the equilibrium curve is determined by the defector's multiplication factor and the group size of the game. Finally, we propose an innovative manifold control approach by incorporating switching control laws which can actually control the stability of the possible manifolds into our general framework. We provide a constructive proof as well as two specific control examples to show the possibility of steering the system states evolving towards the designed directions and entering into any desired region, as long as the initial fraction of cooperators is not too small.

While some previous works have studied the stochastic dynamics on slow manifolds in deterministic dynamical systems [56,57], here, for the first time, we find the emergence of stable equilibrium manifolds in feedback-evolving games. In line with the experimental results, our model reveals great potential to explain and understand the eco-evolutionary dynamics in a variety of real microbial systems. Furthermore, the manifold control method that can steer eco-evolutionary dynamics with external switching feedback control laws fills the challenging theoretical gap in microbial experiments, which may have wide applications in systems biology and microbial ecology.

It is worth noting that the equilibrium manifold in our model naturally extends the concept of population equilibrium points to a manifold (i.e. curve) of stable equilibria; the ultimate population state will be determined by the initial condition and the eco-evolutionary dynamics with environmental feedback. This mathematical extension is not only biologically plausible, but also enables the combination of control theory and evolutionary game theory that aims to steer the population to the desired states. As the observed coevoluntionary behaviours are largely determined by the nonlinear asymmetrical feedback which is related to the selection gradient, we may expect qualitatively similar behaviours in other kinds of social dilemma games as well [58]. Although our present study is focused on group cooperation in multi-person PGGs, our results on manifold control are applicable to many other important situations, such as balancing excitatory and inhibitory interactions in neuronal populations and suppressing evolution of drug resistance in cancer treatment, just to name a few. Finally, since our framework is performed in

the well-mixed population, the evolutionary results in the structured population could be slightly different, as observed in some microbial experiments, which may need further studies [59].

Data accessibility. The experimental data and the source codes for reproducing the separatrix line, phase graphs and control examples are available at https://github.com/fufeng/Manifold-control-for-coevolutionary-games.

Authors' contributions. X.W. and F.F. conceived the model. X.W., Z.Z. and F.F. performed the data and theoretical analyses and wrote the manuscript.

Competing interests. We declare we have no competing interests.

Funding. This work is supported by the Program of National Natural Science Foundation of China (grant nos. 11871004 and 11922102). F.F. is supported by a Junior Faculty Fellowship awarded by the Dean of the Faculty at Dartmouth and also by the Bill & Melinda Gates Foundation (award no. OPP1217336), the NIH COBRE Program (grant no. 1P20GM130454), a Neukom CompX Faculty Grant, the Dartmouth Faculty Startup Fund and the Walter & Constance Burke Research Initiation Award.

Acknowledgements. X.W., Z.Z. and F.F. gratefully acknowledge Alvaro Sanchez, who generously shared the experimental data for our work.

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
