## [Reviewer comments · Proceedings. Mathematical, Physical, and Engineering Sciences]

Review History

RSPA-2019-0643.R0 (Original submission)

Review form: Referee 1

Can the paper be shortened without overall detriment to the main message?

Yes

Do you think some of the material would be more appropriate as an electronic appendix?

No

Do you have any ethical concerns with this paper?

No

Recommendation?

Accept as is

Comments to the Author(s)

In this excellent work the authors suggest a framework which extends the two-player games with environmental feedback to coevolutionary multi-player games with asymmetrical feedback driven by nonlinear selection gradient. As a result, specific effects of asymmetrical feedback loop

driven by nonlinear selection gradient in public goods game are identified. Furthermore, a so-called manifold control method is proposed that can influence the coevolutionary games successfully.

The study of coevolutionary games has been proven to be an adequate tool to model realistic systems [see e.g. *BioSystems* 99 (2010) 109, *EPL* 108 (2014) 28004 or *Sci. Rep.* 6 (2016) 21104]. Present work fits nicely into this research avenue and I found the authors' results not simply sound, but really stimulating. I am quite sure that this paper will collect significant interest from the research community. Therefore I am happy to recommend publication in its present form in *Proceedings A* because the applied technique and the general validity of observed results justify it.

In the case, however, if Editor decides for revision, I would like to give some optional comments. Needless to say, they serve just to elevate the presentation and might help readers to think further some possible consequences of present observations. (But of course, the authors should not necessarily agree with me.) In particular, present study focuses on replicator dynamics that is frequently applied technique for some microbial systems. But this assumption, namely the well-mixed population, is not always valid. See, just a recent experimental work of *Science* 365 (2019) 1045-1049 where the limited interactions of bacterias is an essential condition to detect the presented patterns. I only mention it because a brief comment would be useful that results in structured populations could be slightly different - as it is frequently highlighted by previous works (there are many). My second note is about the possible oscillation due to coevolutionary dynamics. Oscillation may be related to spreading (or invasion) process. But the latter could be qualitatively different for some social dilemma games (as it was recently pointed out in *Sci. Rep.* 9 (2019) 12575). Therefore a brief discussion would be welcomed whether the presented results are limited to public goods games or qualitatively similar behavior are expected for other kind of social dilemmas.

Review form: Referee 2

Can the paper be shortened without overall detriment to the main message?

Yes

Do you think some of the material would be more appropriate as an electronic appendix?

No

Do you have any ethical concerns with this paper?

No

Recommendation?

Accept with minor revision (please list in comments)

Comments to the Author(s)

In this manuscript, the authors provide a modeling framework in multi-player games with asymmetrical feedback driven by nonlinear selection gradient. They find that a larger relative asymmetrical feedback speed for group interactions centered on cooperators not only accelerates the convergence of stable manifolds, but also increases the attraction basin of these stable manifolds. They offers an innovative manifold control approach to steer the eco-evolutionary dynamics to any desired population states. I have enjoyed reading this paper, and I have found it interesting. The results are convincing and give significant values to the readers. Therefore, I am

in favor of publication of this manuscript in Proceedings of the Royal Society A. However, I just have one comment for the authors: In this manuscript, the authors stress that the general form of nonlinear control function can be described by Eq. (2.4). However, it is not clear to me. I wonder why there is no $\$x\$$ term in the general control function. I suggest the authors give some more explanations for Eq. (2.4).

Decision letter (RSPA-2019-0643.R0)

11-Nov-2019

Dear Dr Wang,

On behalf of the Editor, I am pleased to inform you that your Manuscript RSPA-2019-0643 entitled "Steering Eco-Evolutionary Game Dynamics with Manifold Control" has been accepted for publication subject to minor revisions in Proceedings A. Please find the referees' comments below.

The reviewer(s) have recommended publication, but also suggest some minor revisions to your manuscript. Therefore, I invite you to respond to the reviewer(s)' comments and revise your manuscript. Please note that we have a strict upper limit of 28 pages for each paper. Please endeavour to incorporate any revisions while keeping the paper within journal limits. Please note that page charges are made on all papers longer than 20 pages. If you cannot pay these charges you must reduce your paper to 20 pages before submitting your revision. Your paper has been ESTIMATED to be 20 pages. We cannot proceed with typesetting your paper without your agreement to meet page charges in full should the paper exceed 20 pages when typeset. If you have any questions, please do get in touch.

It is a condition of publication that you submit the revised version of your manuscript within 7 days. If you do not think you will be able to meet this date please let me know in advance of the due date.

To revise your manuscript, log into <https://mc.manuscriptcentral.com/prsa> and enter your Author Centre, where you will find your manuscript title listed under "Manuscripts with Decisions." Under "Actions," click on "Create a Revision." Your manuscript number has been appended to denote a revision.

You will be unable to make your revisions on the originally submitted version of the manuscript. Instead, revise your manuscript and upload a new version through your Author Centre.

IMPORTANT: Your original files are available to you when you upload your revised manuscript. Please delete any redundant files before completing the submission process.

In addition to addressing all of the reviewers' and editor's comments, your revised manuscript

MUST contain the following sections before the reference list (for any heading that does not apply to your work, please include a comment to this effect):

- Acknowledgements
- Funding statement

See <https://royalsociety.org/journals/authors/author-guidelines/> for further details.

When uploading your revised files, please make sure that you include the following as we cannot proceed without these:

- 1) A text file of the manuscript (doc, txt, rtf or tex), including the references, tables (including captions) and figure captions. Please remove any tracked changes from the text before submission. PDF files are not an accepted format for the "Main Document".
- 2) A separate electronic file of each figure (tif, eps or print-quality pdf preferred). The format should be produced directly from original creation package, or original software format.
- 3) Electronic Supplementary Material (ESM): all supplementary materials accompanying an accepted article will be treated as in their final form. Note that the Royal Society will not edit or typeset supplementary material and it will be hosted as provided. Please ensure that the supplementary material includes the paper details where possible (authors, article title, journal name). Supplementary files will be published alongside the paper on the journal website and posted on the online figshare repository (<https://figshare.com>). The heading and legend provided for each supplementary file during the submission process will be used to create the figshare page, so please ensure these are accurate and informative so that your files can be found in searches. Files on figshare will be made available approximately one week before the accompanying article so that the supplementary material can be attributed a unique DOI.

Alternatively you may upload a zip folder containing all source files for your manuscript as described above with a PDF as your "Main Document". This should be the full paper as it appears when compiled from the individual files supplied in the zip folder.

Article Funder

Please ensure you fill in the Article Funder question on page 2 to ensure the correct data is collected for FundRef (<http://www.crossref.org/fundref/>).

Media summary

Please ensure you include a short non-technical summary (up to 100 words) of the key findings/importance of your paper. This will be used for to promote your work and marketing purposes (e.g. press releases). The summary should be prepared using the following guidelines:

- *Write simple English: this is intended for the general public. Please explain any essential technical terms in a short and simple manner.
- *Describe (a) the study (b) its key findings and (c) its implications.
- *State why this work is newsworthy, be concise and do not overstate (true 'breakthroughs' are a rarity).
- *Ensure that you include valid contact details for the lead author (institutional address, email address, telephone number).

Cover images

We welcome submissions of images for possible use on the cover of Proceedings A. Images should be square in dimension and please ensure that you obtain all relevant copyright permissions before submitting the image to us. If you would like to submit an image for consideration please send your image to proceedingsa@royalsociety.org

Once again, thank you for submitting your manuscript to Proceedings A and I look forward to receiving your revision. If you have any questions at all, please do not hesitate to get in touch.

Best wishes
Raminder Shergill
proceedingsa@royalsociety.org
Proceedings A

on behalf of
Professor Matjaz Perc
Board Member
Proceedings A

Reviewer(s)' Comments to Author:

Referee: 1

Comments to the Author(s)

In this excellent work the authors suggest a framework which extends the two-player games with environmental feedback to coevolutionary multi-player games with asymmetrical feedback driven by nonlinear selection gradient. As a result, specific effects of asymmetrical feedback loop driven by nonlinear selection gradient in public goods game are identified. Furthermore, a so-called manifold control method is proposed that can influence the coevolutionary games successfully.

The study of coevolutionary games has been proven to be an adequate tool to model realistic systems [see e.g. *BioSystems* 99 (2010) 109, *EPL* 108 (2014) 28004 or *Sci. Rep.* 6 (2016) 21104]. Present work fits nicely into this research avenue and I found the authors' results not simply sound, but really stimulating. I am quite sure that this paper will collect significant interest from the research community. Therefore I am happy to recommend publication in its present form in Proceedings A because the applied technique and the general validity of observed results justify it.

In the case, however, if Editor decides for revision, I would like to give some optional comments. Needless to say, they serve just to elevate the presentation and might help readers to think further some possible consequences of present observations. (But of course, the authors should not necessarily agree with me.) In particular, present study focuses on replicator dynamics that is frequently applied technique for some microbial systems. But this assumption, namely the well-mixed population, is not always valid. See, just a recent experimental work of *Science* 365 (2019) 1045-1049 where the limited interactions of bacterias is an essential condition to detect the presented patterns. I only mention it because a brief comment would be useful that results in structured populations could be slightly different - as it is frequently highlighted by previous works (there are many). My second note is about the possible oscillation due to coevolutionary dynamics. Oscillation may be related to spreading (or invasion) process. But the latter could be

qualitatively different for some social dilemma games (as it was recently pointed out in Sci. Rep. 9 (2019) 12575). Therefore a brief discussion would be welcomed whether the presented results are limited to public goods games or qualitatively similar behavior are expected for other kind of social dilemmas.

Referee: 2

Comments to the Author(s)

In this manuscript, the authors provide a modeling framework in multi-player games with asymmetrical feedback driven by nonlinear selection gradient. They find that a larger relative asymmetrical feedback speed for group interactions centered on cooperators not only accelerates the convergence of stable manifolds, but also increases the attraction basin of these stable manifolds. They offers an innovative manifold control approach to steer the eco-evolutionary dynamics to any desired population states. I have enjoyed reading this paper, and I have found it interesting. The results are convincing and give significant values to the readers. Therefore, I am in favor of publication of this manuscript in Proceedings of the Royal Society A. However, I just have one comment for the authors: In this manuscript, the authors stress that the general form of nonlinear control function can be described by Eq. (2.4). However, it is not clear to me. I wonder why there is no x term in the general control function. I suggest the authors give some more explanations for Eq. (2.4).

Author's Response to Decision Letter for (RSPA-2019-0643.R0)

See Appendix A.

Decision letter (RSPA-2019-0643.R1)

18-Nov-2019

Dear Dr Wang

I am pleased to inform you that your manuscript entitled "Steering Eco-Evolutionary Game Dynamics with Manifold Control" has been accepted in its final form for publication in Proceedings A.

Our Production Office will be in contact with you in due course. You can expect to receive a proof of your article soon. Please contact the office to let us know if you are likely to be away from e-mail in the near future. If you do not notify us and comments are not received within 5 days of sending the proof, we may publish the paper as it stands.

Open access

You are invited to opt for open access, our author pays publishing model. Payment of open access fees will enable your article to be made freely available via the Royal Society website as soon as it is ready for publication. For more information about open access please visit

http://royalsocietypublishing.org/site/authors/open_access.xhtml. The open access fee for this journal is £1700/\$2380/€2040 per article. VAT will be charged where applicable.

Note that if you have opted for open access then payment will be required before the article is published – payment instructions will follow shortly.

If you wish to opt for open access then please inform the editorial office (proceedingsa@royalsociety.org) as soon as possible.

Your article has been estimated as being 21 pages long. Our Production Office will inform you of the exact length at the proof stage.

Proceedings A levies charges for articles which exceed 20 printed pages. (based upon approximately 540 words or 2 figures per page). Articles exceeding this limit will incur page charges of £150 per page or part page, plus VAT (where applicable).

Under the terms of our licence to publish you may post the author generated postprint (ie. your accepted version not the final typeset version) of your manuscript at any time and this can be made freely available. Postprints can be deposited on a personal or institutional website, or a recognised server/repository. Please note however, that the reporting of postprints is subject to a media embargo, and that the status the manuscript should be made clear. Upon publication of the definitive version on the publisher's site, full details and a link should be added.

You can cite the article in advance of publication using its DOI. The DOI will take the form: 10.1098/rspa.XXXX.YYYY, where XXXX and YYYY are the last 8 digits of your manuscript number (eg. if your manuscript number is RSPA-2017-1234 the DOI would be 10.1098/rspa.2017.1234).

For tips on promoting your accepted paper see our blog post: <https://blogs.royalsociety.org/publishing/promoting-your-latest-paper-and-tracking-your-results/>

On behalf of the Editor of Proceedings A, we look forward to your continued contributions to the Journal.

Sincerely,

Raminder Shergill
proceedingsa@royalsociety.org

on behalf of
Professor Matjaz Perc
Board Member
Proceedings A

Appendix A

Response to Comments for Manuscript RSPA-2019-0643

Dear Editor and Referees:

We appreciate referees' careful, professional and efficient review of our manuscript and their valuable suggestions. The comments are encouraging and helpful for improving our paper. We have made all changes exactly according to their suggestions. The detailed responses to referees' comments are listed as follows:

Responses to referees:

Referee 1:

Comments to the Author(s)

In this excellent work the authors suggest a framework which extends the two-player games with environmental feedback to coevolutionary multi-player games with asymmetrical feedback driven by nonlinear selection gradient. As a result, specific effects of asymmetrical feedback loop driven by nonlinear selection gradient in public goods game are identified. Furthermore, a so-called manifold control method is proposed that can influence the coevolutionary games successfully.

The study of coevolutionary games has been proven to be an adequate tool to model realistic systems [see e.g. BioSystems 99 (2010) 109, EPL 108 (2014) 28004 or Sci. Rep. 6 (2016) 21104]. Present work fits nicely into this research avenue and I found the authors' results not simply sound, but really stimulating. I am quite sure that this paper will collect significant interest from the research community. Therefore I am happy to recommend publication in its present form in Proceedings A because the applied technique and the general validity of observed results justify it.

Thank you: we thank the Referee for his/her encouraging comments as well as the helpful suggestions which help improve the scientific rigor of our work. We are happy that the Referee likes our contributions, and hope we have amended the manuscript to his/her full satisfaction.

1. In the case, however, if Editor decides for revision, I would like to give some optional comments. Needless to say, they serve just to elevate the presentation and might help readers to think further some possible consequences of present observations. (But of course, the authors should not necessarily agree with me.) In particular, present study focuses on replicator dynamics that is frequently applied technique for some microbial systems. But this assumption, namely the well-mixed population, is not always valid. See, just a recent experimental work of Science 365 (2019) 1045-1049 where the limited interactions of bacterias is an essential condition to detect the presented patterns. I only mention it because a brief comment would be useful that results in structured populations could be slightly different - as it is frequently highlighted by previous works (there are many). My second note is about the possible oscillation due to coevolutionary dynamics. Oscillation may be related to

spreading (or invasion) process. But the latter could be qualitatively different for some social dilemma games (as it was recently pointed out in Sci. Rep. 9 (2019) 12575). Therefore a brief discussion would be welcomed whether the presented results are limited to public goods games or qualitatively similar behavior are expected for other kind of social dilemmas.

Response: We appreciate these constructive suggestions. We have added some brief comments into the last paragraph of the Discussions part to discuss the limitations (well-mixed population) and the potential extensions (to other kind of social dilemmas) of our current work, with the two recommended papers being cited:

“As the observed coevolutionary behaviors are largely determined by the nonlinear asymmetrical feedback which is related to the selection gradient, we may expect qualitatively similar behaviors in other kinds of social dilemma games as well [58].”

“Finally, since our framework is performed in the well-mixed population, the evolutionary results in the structured population could be slightly different as observed by some microbial experiments, which may need further studies [59].”

Besides, we have also added the suggested references ([53-55]) at the beginning of the Conclusions and Discussions part to support the description of previous studies of the coevolutionary games, which are listed as follows:

- Perc, M., & Szolnoki, A. (2010). Coevolutionary games—a mini review. *BioSystems*, 99(2), 109-125.
- Szolnoki, A., & Perc, M. (2014). Coevolutionary success-driven multigames. *EPL (Europhysics Letters)*, 108(2), 28004.
- Li, K., Szolnoki, A., Cong, R., & Wang, L. (2016). The coevolution of overconfidence and bluffing in the resource competition game. *Scientific reports*, 6, 21104.

Referee 2:

Comments to the Author(s)

In this manuscript, the authors provide a modeling framework in multi-player games with asymmetrical feedback driven by nonlinear selection gradient. They find that a larger relative asymmetrical feedback speed for group interactions centered on cooperators not only accelerates the convergence of stable manifolds, but also increases the attraction basin of these stable manifolds. They offers an innovative manifold control approach to steer the eco-evolutionary dynamics to any desired population states. I have enjoyed reading this paper, and I have found it interesting. The results are convincing and give significant values to the readers. Therefore, I am in favor of publication of this manuscript in Proceedings of the Royal Society A.

Thank you: we are grateful for the Referee’s support. We are glad that the Referee enjoyed our paper, and hope we have amended the manuscript to his/her full satisfaction.

1. However, I just have one comment for the authors: In this manuscript, the authors stress that the general form of nonlinear control function can be described by Eq. (2.4). However, it is not clear to me. I wonder why there is no x term in the general control function. I suggest the authors give some more explanations for Eq. (2.4).

Response: Sorry for the unclear expressions in Eq. (2.4). In fact, the x term is implicit and is included in the terms of P_c and P_d . To make it clear, we have explicitly added the parameter information into Eq. (2.5):

$$\phi_0(x, r_c) = P_c(x, r_c) - P_d(x, r_d)$$

$$\phi_i(x, r_c) = \theta_i P_c(x, r_c) - P_d(x, r_d).$$

We hope that these modifications will meet with approval. Once again, thank you very much for the excellent and professional comments and suggestions of our manuscript.

Best regards,

Feng Fu, Ph.D. (also on behalf of Xin Wang & Zhiming Zheng)